# Moving toward Appropriate Motor Assessment Tools in People Affected by Severe Acquired Brain Injury: A Scoping Review with Clinical Advices

**DOI:** 10.3390/healthcare10061115

**Published:** 2022-06-15

**Authors:** Mirjam Bonanno, Rosaria De Luca, William Torregrossa, Paolo Tonin, Rocco Salvatore Calabrò

**Affiliations:** 1Neurorehabilitation Unit, IRCCS Centro Neurolesi “Bonino-Pulejo”, 98121 Messina, Italy; mirjam.bonanno@irccsme.it (M.B.); rosaria.deluca@irccsme.it (R.D.L.); william.torregrossa@irccsme.it (W.T.); 2Sant’Anna Institute, 88900 Crotone, Italy; patonin18@gmail.com

**Keywords:** severe acquired brain injury, motor assessment, outcomes, disorder of consciousness

## Abstract

Severe acquired brain injury (SABI) is among the leading causes of death and disability worldwide. Patients following SABI may develop motor, sensory and cognitive disorders, alone or in combination. This review aims to point out the most used scales to assess motor function in SABI patients, also attempting to give some indications on their applicability in clinical practice. Studies were identified by searching on PubMed, Web of Science, PeDro and Cochrane databases between January and March 2022. We found that motor assessment tools are currently used by researchers/clinicians either in the acute/post-acute phase (for prognosis and rehabilitation purposes) or in the chronic phase (when functional items may also be considered). Moreover, specific scales exist only for patients with disorders of consciousness, whereas regarding motor function, SABI is mainly assessed by adapting the tools commonly used for stroke. Although some doubts remain about the validity of some of these assessment tools in SABI, to investigate motor outcomes is fundamental to establish a correct prognosis and plan a tailored rehabilitation training in these very frail and vulnerable patients.

## 1. Introduction

Severe acquired brain injury (SABI) is one of the leading causes of death and disability worldwide, and this is mainly caused by head trauma or ischemic and hemorrhagic vascular lesions. To be diagnosed with SABI, a patient must be in a state of coma for at least 24 h [1,2,3]. A recent review shows that in Italy, traumatic BI incidence is 849 cases per 100,000 inhabitants with a mean age of 26.7–44.5 and higher incidence in males [3]. Usually, SABI results in long-term motor, sensory and cognitive–behavioral problems [1,2]. However, chronic disorder of consciousness (DoC), i.e., vegetative state (VS) and minimally conscious state (MCS), can occur. Nowadays, the term vegetative state has been replaced by the “unresponsive wakefulness syndrome” (UWS). VS/UWS involves the lack of interaction with the surrounding environment; patients do not show self-awareness and cannot interact with other people. In MCS, unlike VS/UW, there is evidence that patients are somehow aware of themselves and/or their environment [4]. Patients also could tend to gradually become more conscious and can go from a VS to MCS, sometimes years after the original brain damage [5]. In addition to problems with consciousness, patients following SABI have specific motor deficits, such as hemiparesis, spasticity in the upper and lower limbs, and loss of trunk control that limit breathing and communication [6,7]. All of these physical disabilities associated with cognitive impairment can cause significant handicaps and limit the return to a normal and productive life [8]. Residual gross motor functions are evaluated through rating scales that monitor, for example, the recovery of simple motor tasks such as the transition from supine to sitting and from sitting to standing, and the acquisition of head and trunk control [9]. In clinical practice, however, it is not always easy to find the right rating scale to assess motor impairment and disability. In fact, although there are different assessment tools that can be used both in traumatized patients and in those with stroke, some of these scales evaluate the autonomy in daily activity but do not evaluate specific aspects of motor recovery. In addition, it is necessary to separate assessment scales that must be administered in the acute phase from those that are administered during the intermediate sub-acute and long-term phase. In acute, the primary evaluation in most studies [10,11,12] is given by the Glasgow Coma Scale. Specifically, it is concerned with assessing arousal and awareness as components of consciousness, reflecting the person’s conscious state [13]. Among the scales that rate outcomes in acute care hospitals, there is the Disability Rating Scale (DRS). The DRS was developed and tested in patients with moderate and severe head injury, especially during rehabilitation [14]. In the following phases, the residual motor functions will be evaluated when the patient is stabilized and consciousness improves.

The purpose of this scoping review is to investigate the most used and appropriate assessment tools for motor recovery in patients with SABI. Moreover, we want to give some clinical indications on the proper use of the different scales along the complex rehabilitation path of these patients.

## 2. Materials and Methods

The data collected were obtained through a search on PubMed, Web of Science, Cochrane databases and PeDro. We considered the range period January 2010–March 2022, because this last decade has witnessed the implementation and development of SABI patients’ management, including motor assessment. We used the combinations of the following words “motor evaluation” AND “severe brain injury” OR “severe stroke” OR “disorder of consciousness” OR “acquired brain injury” OR “movement evaluation” OR “assessment scales” OR “Glasgow coma scale”. In order to obtain a complete search, we have also analyzed the references of the selected articles. We screened 200 studies, and among these, 36 studies were initially selected, and eventually 26 were included and analyzed, as they met the inclusion criteria (Figure 1).

### Selection Study

Inclusion criteria were: (i) adult patients (age >/= 18), (ii) patients with severe stroke and severe TBI, (iii) other severe brain lesions that may cause a state of coma for at least 24 h; and (iv) a GCS score between 3 and 8. The exclusion criteria were: (i) patients with minor to moderate stroke and TBI; (ii) patients with previous psychiatric disorders (psychosis, delirium, depression and (iii) previous motor impairments.

## 3. Results

We analyzed the most used motor rating scales by researcher and clinicians, according to the level of disability (Table 1). In fact, it is necessary to distinguish between scales administered in acute/post-acute and chronic phases as well as in patients with severe and mild to moderate disability. Notably, we found a higher number of specific scales for motor activity in the upper limb rather than in control of the head, trunk and gait.

Moreover, the psychometric properties of motor rating scales (i.e., Reliability, Validity and Responsiveness/Sensitivity) represent the degree of precision with which a test allows to assess a certain ability. In Appendix A, we analyzed the properties of the most used motor rating scales.

### 3.1. Disorder of Consciousness Evaluation

The initial assessment involves patients that are more severely affected, i.e., when the SABI patient is still in a phase of a VS/UWS or MCS. We found four articles describing the use of the Glasgow Coma Scale (GCS), the Glasgow Outcome Scale Extended, Coma Recovery Scale revised (CRS-r), and the Functional State Examination (Table 1). In fact, according to the literature, these are the most used assessment tools of SABI patients either in the acute or post-acute phases. Notably, the GCS is the most used scale to assess the level of consciousness [10,11,12,13], while the CRS-R seems underestimated by clinicians, although it is a valid tool for the rehabilitation prognosis in DoC [15]. Among these scales, FSE appears to be more closely related to the severity of the brain injury than GOSE [16,17] (see Table 2).

### 3.2. Trunk Control

Trunk control was assessed in five studies, mainly using the following three scales: Trunk Recovery Scale (TRS), Trunk Control Test (TCT) and Trunk Impairment Scale (TIS). Among these, TIS was considered by Monticone et al. [19] as an excellent tool for patients with both acute and chronic stroke, while TRS and TCT have been administered both in patients suffering from stroke and traumatic brain injury. In particular, in the study by Montecchi et al. [20], TCT had an excellent predictive validity when associated with TRS in patients with TBI.

### 3.3. Lower Limb Motor Activity

The scales that evaluate the functionality of the lower limb in patients with SABI are very few; in fact, we found only three studies. The study by Hernandez et al. [21] considered FMA-LE as the gold standard for lower limb activity, especially in post-stroke patients. Belluscio et al. [22] assessed the walking quality of patients with TBI using the 10-min walking test, suggesting the adequacy of an integrated assessment using both clinical scales and wearable sensors to objectively assess gait. Finally, Sommerfeld et al. [23] used RMI as a prognostic factor in older patients following severe stroke in association with the Barthel Index (BI) (Table 3).

### 3.4. Upper Limb Motor Activity

The most used assessment tool for the upper limb was the Fugl–Meyer Assessment for Upper Limb (FMA-UL), both for its optimal psychometric properties (reliability, validity and responsiveness) and the potential to evaluate motor skills, coordination, and reflexes of the upper limb in SABI patients [24,25]. On the other hand, some scales for motor dexterity in the activity of daily living, such as the Action Research Arm Test (ARAT) [26], the Chedoke Arm and Hand Inventory (CAHAI) [27], the Box and Block Test (BBT) [28] and Wolf Motor Function Test (WMFT) [29,30], have been used also by some authors in post-stroke patient population. In particular, among the studies selected, only Johnson et al. [27] administered the Chedoke Arm and Hand Inventory (CAHAI) in a sample of patients with acquired brain injury, measuring its reliability among the evaluators (Table 4).

### 3.5. Spasticity Assessment

Among the assessment tools for spasticity, only two main scales, namely the Modified Ashworth Scale and Tardieu Scale, were used. The MAS is definitely the scale that until now is mostly used by clinicians for the measurement of spasticity due to its simplicity and shorter application time. Nonetheless, besides MAS and TS, Marciniack et al. [32] have suggested using some other evaluation scales, including the Spasm Frequency Scale, FIM and the FMA, to better and more objectively assess spasticity. While MAS and TS have been indicated as the only measures in post-stroke patients [33], Thibaut et al. [34] exclusively used MAS for the assessment of spasticity in disorders of consciousness. The main difference between MAS and TS consists in evaluating different aspects of spasticity. In fact, MAS allows the assessment of the degree of spasticity through muscular resistance to passive movement, while TS tests muscular responses of the moving limb recorded at three different speeds (as slow as possible = V1; rate of fall of the segment subjected to gravity = V2; as fast as possible = V3) [32,33,34].

### 3.6. Global Functioning Assessment

We found only four studies dealing with the global functioning of the patient with SABI. Deepika et al. [14] demonstrated that DRS is useful in predicting an unfavorable outcome at six months from the discharge of the patients with SABI. The global functioning of the person with SABI can be evaluated, according to the authors, with the AMPS [35], which highlights the participation of the ADL patient and with the BI-FOM [36]. However, none of these authors has taken into consideration, in addition to the overall functioning of the subject, even the neurological one. For these reasons, Wilde et al. [37] have formulated an ad hoc version of the already validated NIHSS, transforming it into Neurological Outcome Scale—TBI, which also considers the neurological consequences after TBI (Table 5).

## 4. Discussion

In rehabilitation medicine, the assessment of the effectiveness of therapeutic interventions must be carried out through the use of measurement scales that are able to detect the possible and expected change (outcome) after the training. This is particularly true in patients with DoC where prognosis is very difficult to establish. In the present review, we have then pointed out the most used scales to assess motor function in these very frail patients, also attempting to give some indications on their applicability in clinical practice. In particular, a recent study by Dikman and colleagues [17] has compared the effectiveness of the FSE and GOSE, demonstrating that both scales are sensitive to establish the severity of brain injury and recovery of function over time. Six months after the injury, FSE appears to be more closely related to the severity of the brain injury than GOSE [17]. However, the lack of effectiveness in understanding the evolution of VS/UWS has highlighted the need for an evaluation scale more sensitive to change of DoC. Indeed, our results show that the CRS-R not only is a valuable diagnostic tool to differentiate between VS/UWS and MCS, but it also determines the prognosis, subsequent rehabilitation and end-of-life choices for patients and their families, thus bearing ethical implications. According to Chaturvedi et al. [15], an improvement on the CRS-R during the first 4 weeks of hospital stay of survivors of SABI was associated with a better outcome at discharge, independently of age, sex, etiology, time post onset, and presence of main clinical complications. Moreover, patients with MCS show more continuous improvement and attain significantly more favorable outcomes by 1 year [15].

Recovery from a DoC, caused both by traumatic and vascular events, is a complex process that occurs through a combination of spontaneous and mediated processes. Although structural and functional cerebral impairment likely recovers through a potentiating and extension of residual brain areas [38], motor recovery can occur [18,39] and involves in a different way the trunk, LE, or UL. In fact, in neurorehabilitation, trunk recovery is one of the main neurological goals, and it needs standardized postural control assessment for SABI patients. However, a few clinical assessment tools (TCT, TIS and TRS) have been developed and standardized to detect postural control recovery after neural damage [40]. TRS could be administered in the acute phase, as the items consider the trunk control in supine and sitting postures, and then, this scale better suits DoC patients. One of the main prognostic factors for motor recovery is reaching the seated position [41], but not all patients with SABI can sit on their own. TCT has the advantage to assess turning in the bed for those patients who cannot sit independently, as evidence of trunk “rolling” can predict motor recovery and comprehensive activities of daily living [42]. On the other hand, according to the study by Monticone et al. [19], TIS is a good tool for the evaluation of the sitting position in patients with both acute and chronic stroke. Then, it would seem more suitable for those patients with SABI who have already partially recovered trunk control, as it assesses the ability to remain in a sitting position with legs crossed and the voluntary lateral flexion of the trunk [43].

Gait recovery is one of the most important goals in the rehabilitation of patients with brain injury [44]. However, unlike the several scales for the upper limb, very few scales were found for the assessment of the lower limb. Hernàndez et al. [21] suggest that FMA-LE represents the gold standard for evaluating lower limb functions in post-stroke patients, also thanks to the excellent intra- and inter-rater reliability in rehabilitation settings. Despite its remarkable features, we did not find any studies using FMA-LE in SABI patients. Nonetheless, we believe that this scale could instead be helpful to clinicians to investigate gross motor functions in this patient population. Notably, 10MWT and F8WT [22], as well as RMI [23], were used to discriminate different levels of gait in patients with mild to moderate acquired brain injury.

Clinicians and researchers need to identify appropriate measures that have sound assessment properties (i.e., reliability, validity, and responsiveness) to determine the effects of UL training on dexterity of the paretic hand [45]. The use of appropriate measures for outcome evaluations would enhance the methodological quality of controlled trials in SABI rehabilitation research. Among the scales for the upper limb, the FMA [46,47,48] has a good reliability, validity and responsiveness [49], although there are some problems with its administration [24]. Hijikata et al. [50] analyzed the difficulty of performing some items of the FMA-UL in patients with both moderate and severe chronic stroke, showing that all coordination/speed items were at the most difficult levels. Chen and colleagues [51] attempted to show which version of the FMA was best for brain injury patients, considering the shorter version (FMA-12 items) and the longer version (FMA-37). The results of the study showed that the 37-item Fugl–Meyer motor scale had better reactivity than the short one due to its better ability to detect changes in UL and LE function; therefore, the authors recommend the long tool version as a measure of outcomes. Interestingly, Villàn-Villàn et al. [25] suggest the use of FMA-UL for the evaluation of SABI patients to achieve a more personalized and potentially effective physical rehabilitation. Among the other scales, WMFT was developed specifically for the evaluation of the effects of a specific kind of therapy such as constriction-induced movement therapy (CIMT) in post-stroke patients [29]. The test (like the others for UL) evaluates the unilateral performance on functional tasks, and it has been used mainly in mild to moderate patients. Indeed, manual recovery of dexterity depends on a large repertoire of functional and structural processes within the central nervous system, known as neuroplasticity, which occur spontaneously or are induced by the practice of repetitive and task-oriented motor activities [52]. For these reasons, according to the reported literature, UL tests for manual dexterity [31,53,54] can be administered only when patients with SABI regain an optimal motor control as well as a discrete cognitive function. Then, in clinical practice, they cannot so often be used.

Another aspect that severely limits the quality of life of patients with SABI is spasticity, which is defined as a speed-dependent increase in muscle tone that is often accompanied by painful spasms. When spasticity is prolonged over time, as it occurs in DoC, it causes a change in the structure of the periarticular tissues and in the shortening of the muscles that lead to reduced mobility both active and passive [34]. The quantification of spasticity remains an unsolved and difficult problem. MAS is a tool widely used in clinical practice, as it is easy and quick to use and is the most used by researchers, whereas TS is instead less applied in neurorehabilitation settings. However, there are conflicting opinions as to which of the two scales is better than the other in assessing spasticity [32], although it has been found that MAS does not adequately assess spasticity [33] when comparing the test–retest reliability and the reliability between the evaluators of the two scales. In fact, it seems that spasticity can cause tissue changes and contractures that cannot be distinguished by only using MAS [55,56].

Assessment of general functioning and health status in daily life remains a major challenge for researchers today, especially in patients with SABI. In particular, the assessment of the quality of life could be carried out in clinical practice when SABI patients become more actively involved in ADL [57]. Toneman et al. evaluated the use of AMPS, a rating scale used for the assessment of participation in ADL, which is standardized, valid, reliable, and sensitive to changes over time [35]. However, in SABI patients’ everyday life, the presence of a caregiver is fundamental to optimize ADL/IADL and favor a better quality of life [58]. In fact, the family members of SABI patients can be useful to potentiate general communication and interpersonal abilities and improve the global cognitive status and sensory–motor outcomes of patients. On the other hand, Deepika et al. [14] used DRS to predict the functional but also unfavorable outcome at the discharge of SABI patients. Another evaluation measure of the person’s overall functioning after a head injury is the BI-FOM to monitor recovery and response to rehabilitation over time [36]. Finally, Wilde et al. found that NOS-TBI is useful to stratify the full spectrum of TBI severity, although it is specific for traumatic head injury patients [37].

It is noteworthy that patients with severe acquired brain injury who are unable to communicate verbally could be assessed using eye-tracking technology especially for cognitive function, particularly when eye movements are the only channel of communication in individuals after brain damage. Notably, these tools limit the patient’s isolation and encourage the autonomy of the individual with DoC and/or SABI, also improving the cognitive outcomes [59,60].

The main limitation of this scoping review is the absence of a quality assessment of the paper [61]. However, we aimed to investigate the main used scales in the assessment of SABI to provide clinicians with clinical advices. A further systematic review overcoming this important consideration is needed in order to better point out the validity of the papers and the reliability of the scales.

## 5. Clinical Advices

Given that various motor assessment tools for patients with SABI exist, researchers as well as clinicians do not always agree on which tool is better for the different post-coma conditions. With our work, we have tried to bridge the gap regarding the motor assessment of these frail individuals during the clinical stages of the disease (acute, post-acute and chronic), which is strictly related to the extent of the brain damage:(1)In the acute phase, clinicians could use GCS to test level of consciousness [11,12] associated to CRS-r [15], which could be helpful to differentiate between UWS/VS and MCS. It also can be useful to determine the prognosis and the subsequent rehabilitation path by using common and more advanced electrophysiological tests. In addition, trunk control could be evaluated early, using the TRS [20], as the items consider the supine and sitting postures, which promote motor recovery.(2)In the post-acute phase, when consciousness and motor activity improve, clinicians could use TCT and/or TIS to assess trunk control [19,20] and FMA-LE to assess lower limb functioning, as this is the best motor outcome measure for the evaluation of lower limb recovery [21];(3)In the chronic phase, in SABI patients with mild to moderate disability, functional ambulation may be tested with 10MWT and F8WT [22], as well as RMI [23], to discriminate different levels of gait. Furthermore, for the upper limb recovery, we suggest to administer FMA-UL [25,26,49] to evaluate the global recovery of upper limb activity because of its psychometric and clinical properties. On the other hand, according to the literature, UL tests for manual dexterity [31,53,54] can be administered only when patients regain an optimal motor control and also a discrete cognitive function. Then, in clinical practice, they cannot be so often used. In addition, global functioning could be evaluated using the DRS, which is a good tool to measure general changes over the course of recovery for individuals with SABI, from coma to return in community [14].

Despite of the evolution of the disease, there are scales for the assessment of specific symptoms, such as spasticity, which can/should be used at any time, i.e., in the acute, post-acute or chronic phase. MAS [32,34] is the most commonly used tool; however, it seems that it fails to properly assess spasticity in the presence of tissue changes and contractures [55,56]. This is why, in these cases, we recommend the combined use of MAS and TS.

Finally, motor outcomes and spasticity may be affected by pain and spasm; then, patients with SABI should be always investigated for these symptoms/signs, although they are out the scope of this review.

## 6. Conclusions

Motor assessment is a crucial point in the management of patients with SABI during both the acute/subacute and chronic phase. The present scoping review has discussed the tools most commonly used by researchers and clinicians in these frail individuals, demonstrating that specific scales exist only in patients with DoC, whereas during the recovery, the scales used to assess stroke are adapted to this patient population. Therefore, future studies should be performed on large samples of SABI patients (both with a vascular and traumatic etiology), encouraging the development of more specific tools for this kind of patients, focusing on lower limb (given that the literature lack detailed data on this issue). Although some doubts remain about the validity of some of these assessment tools in SABI, to investigate motor outcomes is fundamental to establish a correct prognosis and plan a tailored rehabilitation training in these very frail and vulnerable patients.

## Figures and Tables

**Figure 1 healthcare-10-01115-f001:**
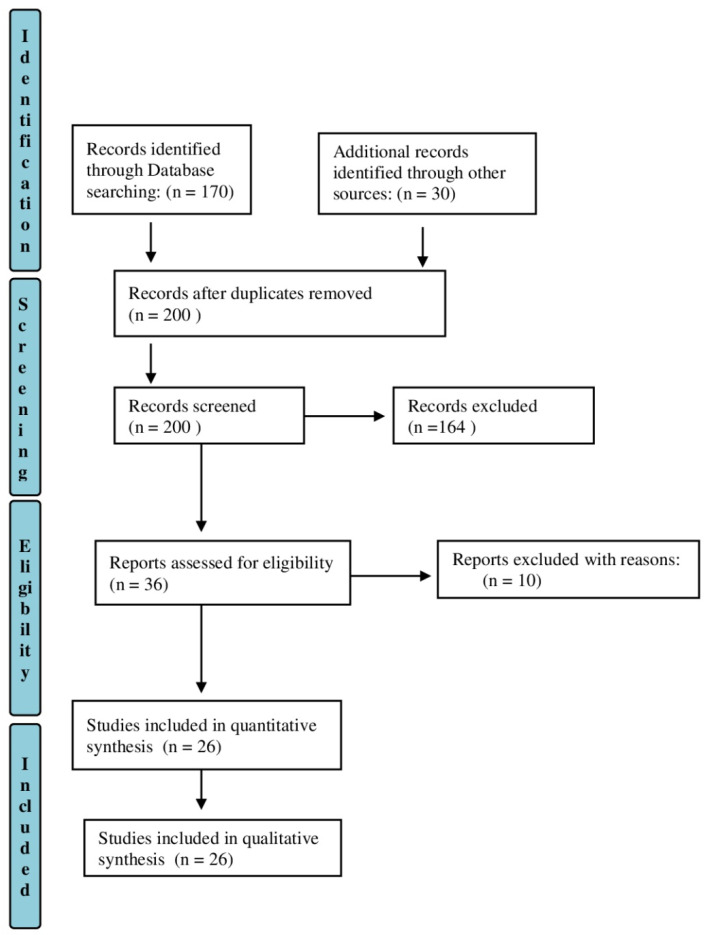
PRISMA flow diagram for study selection.

**Table 1 healthcare-10-01115-t001:** Illustrates the scales assessed in each of the articles selected for this work.

Scales	Research Articles																						
Study Reference *n*	11–12	14	15	16	17	18	19	20	21	22	23	24	25	26	27	28	30–31	32	33	34	35	48–49	53
GCS	X																						
CSR-r			X																				
FSE				X	X																		
GOSE					X																		
TIS						X																	
TRS							X																
TCT							X																
FMA-LE								X															
10MWT									X														
F8WT									X														
RMI										X													
FMA-UL											X	X										X	
ARAT													X										
AMAT																							X
CAHAI														X									
BBT															X								
WMFT															X	X							
MAS																	X	X					
TS																	X	X					
DRS		X																					
AMPS																			X				
BI-FOM																				X			
NOS-TBI																					X		

X = the study has assessed the specific scale variable. Legend: GCS: Glasgow Coma Scale; CSR-r: Coma Recovery Scale—revised; FSE: Functional State Examination; GOSE: Glasgow Outcome Scale Extended; TIS: Trunk Impairment Scale; TRS: Trunk Recovery Scale; TCT: Trunk Control Test; FMA-LE: Fugl–Meyer Assessment—Lower Extremity; 10MWT: 10 Minute Walking Test; F8WT: The Figure of 8 Walking Test; RMI: Rivermead Mobility Index; FMA-UL: Fugl–Meyer—Upper Limb; ARAT: Action Research Arm Test; AMAT: Arm Motor Ability Inventory; BBT: Box and Block Test; CAHAI: Chedoke Arm and Hand Activity Inventory; WMFT: Wolf Motor Function Test; MAS: Modified Ashworth Scale; TS: Tardieu Scale; DRS: Disability Rating Scale; AMPS: Assessment of Motor and Process Skills; BI-FOM: Brain Injury Functional Outcome Measure.

**Table 2 healthcare-10-01115-t002:** Assessment consciousness scales to use during the vegetative state or minimally conscious state.

Scale	Description
GCS (Rozenfeld et al., 2020 [11], McKee et al., 2015 [18])	The GCS is a tool used to assess and calculate a patient’s level of consciousness. Teasdale and Jennett presented, for the first time, the GCS in 1974 as an aid in the clinical assessment of unconsciousness The GCS uses a three-criteria scoring system: best eye opening (maximum 4 points), best verbal response (maximum 5 points), and best motor response (maximum 6 points). These scores are added together to give a total score between 3 and 15.
GOSE (Dikmen et al., 2019 [17])	The Extended Glasgow Outcome Scale (GOS-E) was created as an advancement from the original GOS. It includes: (1) Death, (2) Vegetative state, (3) Lower severe disability, (4) Superior severe disability, (5) Lower moderate disability, (6) Upper moderate disability—some disability but may exist in part to resume work, (7) Lower to good healing—minor physical or mental deficits, (8) Superior good recovery—full recovery.
CRS-r (Chaturvedi et al., 2021 [15])	The coma recovery (CRS-r) can be used to differentiate DoC; it consists of 23 items divided into six subscales designed to assess brain functional ability for auditory, visual, motor, verbal, communication, and arousal functions.
FSE (Machamer et al., 2018 [16]; Dikmen et al., 2019 [17])	The Functional Status Examination (FSE) is a new measure designed to evaluate change in activities of everyday life as a function of an event or illness, including traumatic brain injury. The measure covers physical, social, and psychological domains.

Legend: GCS: Glasgow Coma Scale; GOSE: Glasgow Outcome Scale Extended; CRS-r: Coma Recovery Scale revised; FSE: Functional State Examination.

**Table 3 healthcare-10-01115-t003:** Description of lower limb assessment scale.

Scale	Description
FMA-LE (Hernandez et al., 2021 [21])	The motor domain includes items assessing movement, coordination, and reflex action of the hip, knee, and ankle. Points are divided among the domains as follows: Motor score: ranges from 0 (hemiplegia) to 100 points (normal motor performance). Divided into 66 points for upper extremity and 34 points for the lower extremity; Sensation: ranges from 0 to 24 points. Divided into 8 points for light touch and 16 points for position sense; Balance: ranges from 0 to 14 points. Divided into 6 points for sitting and 8 points for standing; Joint range of motion: ranges from 0 to 44 points; Joint pain: ranges from 0 to 44 points.
10MWT (Belluscio et al., 2019 [22])	The 10 Meter Walk Test is a performance measure used to assess walking speed in meters per second over a short distance. It can be employed to determine functional mobility, gait, and vestibular function. The individual walks without assistance for 10 m, with the time measured for the intermediate 6 m to allow for acceleration and deceleration. Assistive devices may be used, but they must be kept consistent and documented for each test. It can be tested at either preferred walking speed or maximum walking speed.
F8WT (Belluscio et al., 2019 [22])	The Figure 8 Walk Test (F8WT) measures the everyday walking ability of older adults with mobility disability. The F8WT tests a participant’s gait in both straight and curved paths. The F8WT uses a path where the participant is asked to walk a figure eight shape around two cones. Scores are recorded in three areas: (1) speed (time for completion), (2) amplitude (number of steps taken), and (3) accuracy or “smoothness”.
RMI (Sommerfeld et al., 2011 [23])	The Rivermead Mobility Index assesses functional mobility in gait, balance and transfers after brain injury. The RMI includes fifteen mobility items: 14 self-reported and 1 direct observation (standing unsupported). The 15 items are hierarchically arranged, and all items are ordered according to ascending difficulty, from “turning over in bed” to “able to run”: Each item is coded 0 or 1. A score of 0 = a ‘no’ response; a score of 1 = a ‘yes’ response. A total score is determined by summing the points allocated for all items.

Legend: FMA-LE: Fugl–Meyer Assessment—Lower Extremity; 10MWT: 10 Meter Walking Test; F8WT: Figure of 8 Walk Test; RMI: Rivermead Mobility Index.

**Table 4 healthcare-10-01115-t004:** Description of upper limb (UL) principles scale.

Scale	Description
FMA-UL (Gladston et al., 2002 [24]; Villàn et al., 2018 [25])	Evaluates changes in motor function in the areas of balance, motor skills, coordination and reflexes. The score of the scale ranges from 0 (the patient does not perform the task), 1 (the task is partially performed) and 2 (the task is performed completely). The total upper extremity score is 66.
ARAT (Chen et al., 2012, [26])	ARAT is a measure to evaluate specific changes in upper limb function through 19 items that assesses the ability of the hemiplegic patient to grasp objects of different size, weight and shape.
AMAT (O’Dell et al., 2013, [31])	Arm Motor Ability Inventory is a quantitative analysis of functional ability in carrying out activities of daily life. It consists of nine tasks that replicate basic ADL with the help of the upper limb and the paretic hand. During the administration of the AMAT, the patient can use the healthy limb as compensation, but this determines a penalty in the score.
BBT (Rand et al., 2018, [28])	The Box and Block Test (BBT) measures unilateral gross manual dexterity of hemiparetic patients. The patient seated at a table in front of a rectangular box divided into two compartments and is asked to transfer as many blocks as possible, one at a time from one compartment to another, in sixty seconds
CAHAI (Johnson et al., 2017 [27])	The Chedoke Arm and Hand Activity Inventory is a functional assessment of the recovering arm and hand after stroke. The original CAHAI includes 13 functional items that involve both upper limbs, and it incorporates a range of movements and grasps that reflect stages of motor recovery following stroke.
WMFT (Edwards et al., 2012 [29])	The Wolf Motor Function Test (WMFT) quantifies upper extremity (UE) motor ability through timed and functional tasks, consisting of 17 items. The first 6 items involve timed functional tasks, items 7 and 14 are measures of strength, and the remaining 9 items consist of analyzing movement quality when completing functional tasks. The items are rated on a 6-point scale.

Legend: FMA-UL: Fugl–Meyer Assessment-Upper Limb; ARAT: Action Research Arm Test; AMAT: Arm Motor Ability Inventory; BBT: Box and Block Test; CAHAI: Chedoke Arm and Hand Activity Inventory; WMFT: Wolf Motor Function Test.

**Table 5 healthcare-10-01115-t005:** Assessment global functioning scales to use in severe to moderate disability.

Scale	Description
DRS (Deepika et al., 2017 [14])	The DRS tracks the recovery of an individual from coma to community and to measure general functional changes over the course of recovery for individuals with SABI. DRS evaluates 8 areas of functioning in 4 categories: Consciousness (eye opening, communication ability, and motor response); Cognitive ability (feeding, toileting, grooming); Dependence on others; Employability. Each area of functioning is rated on a scale of 0 to either 3 or 5 (maximum score = 29—vegetative state; minimum score = 0—person without disability)
AMPS (Toneman et al., 2010 [35])	The Assessment of Motor and Process Skills is an observational assessment that measures the performance quality of the tasks related to ADL. The AMPS is designed to examine interplay between the person, the ADL task, and the environment. It includes 16 motor ADL items divided into 4 domains (body positions, obtaining and holding objects, moving self and objects, sustaining performance). The other part includes 20 ADL process skills divided into 5 domains (sustaining performance, applying knowledge, temporal organization, organizing space and objects, adapting performance). Items level scores range from 1 = no problem to 6 = cannot test.
BI-FOM (Whyte et al., 2021 [36])	The BI-FOM improves the global measurement of function after moderate–severe brain injury. The BI-FOM is a composite of 31 items drawn from other measures of functioning.
NOS-TBI (Wilde et al., 2010 [37])	This tool based on the National Institute of Health Stroke Scale, and it stratifies injury severity. It is divided into 23 items, addressing exam elements of orientation, cranial nerve function, strength, sensation, language and coordination. Items are rated on 3–4 or 5 level scales. Higher scores reflect greater neurological impairments.

Legend: DRS: Disability Rating Scale; AMPS: Assessment of Motor and Process Skills; BI-FOM: Brain Injury Functional Outcome Measure.

## Data Availability

Not applicable.

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
