# Peer review of "Moving toward Appropriate Motor Assessment Tools in People Affected by Severe Acquired Brain Injury: A Scoping Review with Clinical Advices"

_healthcare, 2022, doi:10.3390/healthcare10061115_

Round 1
Reviewer 1 Report
This paper has a great purpose, and it will help clinicians and researchers to choose appropriate assessment tools for this population.
- Line 60. The purpose of the study is "to investigate the most used and reliable assessment tools for motor recovery in patients with SABI." What process did you go through to validate "reliable assessment"?
- Line 82-84. Based on your review, can you also provide a list of assessment tools for the different stages of recovery (e.g., acute/post-acute vs. chronic SABI) and the severity of the injury (e.g., severe vs. mild-moderate SABI) in your result section? These were mentioned randomly in some assessments (for example, lines 102-103); however, what assessments were most used for acute, severe SABI patients, for instance?
- For discussion or conclusion, what future studies (e.g., to test the validity of some of these studies, lines 260-261) would be needed to have more standardized assessments for this population?
Author Response
This paper has a great purpose, and it will help clinicians and researchers to choose appropriate assessment tools for this population.
- Line 60. The purpose of the study is "to investigate the most used and reliable assessment tools for motor recovery in patients with SABI." What process did you go through to validate "reliable assessment"?
We have changed reliable into appropriate, to avoid misunderstanding. (line 62)
- Line 82-84. Based on your review, can you also provide a list of assessment tools for the different stages of recovery (e.g., acute/post-acute vs. chronic SABI) and the severity of the injury (e.g., severe vs. mild-moderate SABI) in your result section? These were mentioned randomly in some assessments (for example, lines 102-103); however, what assessments were most used for acute, severe SABI patients, for instance?
This issue has been better specified in the results and at the end with the clinical advices, as suggested.
- For discussion or conclusion, what future studies (e.g., to test the validity of some of these studies, lines 260-261) would be needed to have more standardized assessments for this population?
We added also this information at the lines 411-414.
Reviewer 2 Report
Authors have performed a scoping review on patients suffering Severe Acquired Brain Injury (SABI). They included 36 out of 200 papers during January 2010 - March 2022 period. The topic is interesting and the whole script is written in proper English.
Follow you will find my comments:
- I hesitate to call this review a scoping review. Scoping review are broader, aiming to filling a gap using different search structures. The authors didn’t provide any reason to why they consider their review a scoping one. in line 60 authors called their review a ‘narrative’ review. Please be persistent with the term.
- Though the topic is interesting but the who script is not well organized and written rather superficially. Not presenting proper tables and graphs make it very difficult to track the references.
- The exclusion-inclusion process is not fully described e.g. how many papers have been chosen from the references. I recommend to use a flowchart to show the process of paper selecting.
- Authors didn’t give a reason why they decided to limit their research time range to January 2010 - March 2022.
- No quality assessment is performed on the papers. (Peters et al, Gait & Posture 31 (2010))
- Tables are confusing. It is not clear when the description for one table ends and the other one starts. Having too many tables make it very difficult to track the papers. I highly recommend to have one large table for scales and another one for the selected paper information, including which papers use which scales.
- The tables 1 and 5 formats are slightly different from tables 2 and 3. In one the Scale column is introduced with references. Please use the same format.
- The authors discussed briefly about validity, reliability and sensitivity while these parameters deserve more attention. I recommend to discuss them independently in according subsections , given which papers mainly discussed these parameters.
- Authors mentioned in the title ‘A Scoping Review with Clinical Advices’. The advices should be brought in the conclusion as a final statement of the authors yet it is missed.
Author Response
Authors have performed a scoping review on patients suffering Severe Acquired Brain Injury (SABI). They included 36 out of 200 papers during January 2010 - March 2022 period. The topic is interesting and the whole script is written in proper English.
Follow you will find my comments:
- I hesitate to call this review a scoping review. Scoping review are broader, aiming to filling a gap using different search structures. The authors didn’t provide any reason to why they consider their review a scoping one. in line 60 authors called their review a ‘narrative’ review. Please be persistent with the term.
We have now better specified the reason why this may be considered a scoping review and we have been consistent with the term. In particular, our study is a scoping review because it identifies the evidence available in the motor evaluation of the SABI patient; it clarifies the main measures of motor assessment in the various phases of the disease, from acute, intermediate and chronic; furthermore, the presence of the flowchart highlights how the research is conducted on this certain topic; finally, it identifies and analyzes a knowledge gap.
- Though the topic is interesting but the who script is not well organized and written rather superficially. Not presenting proper tables and graphs make it very difficult to track the references.
Tables have been improved, as suggested.
- The exclusion-inclusion process is not fully described e.g. how many papers have been chosen from the references. I recommend to use a flowchart to show the process of paper selecting.
A flow chart has been added, also to better fit the paper within a scoping review (Figure 1).
- Authors didn’t give a reason why they decided to limit their research time range to January 2010 - March 2022.
In This last decade there was a growing interest in Doc management, as specified (lines 70-71).
- No quality assessment is performed on the papers.
No quality assessment was made (and we found that in many other published scoping review this lacks); However, we have added this as a further limitation of the study.
- Tables are confusing. It is not clear when the description for one table ends and the other one starts. Having too many tables make it very difficult to track the papers. I highly recommend to have one large table for scales and another one for the selected paper information, including which papers use which scales.
We have removed table number 4, and integrated the main informations into the text (lines 221-225); we have also added one large table with all the papers to track better the references (table n°1 in results).
- The tables 1 and 5 formats are slightly different from tables 2 and 3. In one the Scale column is introduced with references. Please use the same format.
As suggested by you, we have organized the tables, describing motor scales, with the same format.
The authors discussed briefly about validity, reliability and sensitivity while these parameters deserve more attention. I recommend to discuss them independently in according subsections, given which papers mainly discussed these parameters.
We have now discussed this important issue in appendix, to let the readers better understand.
Authors mentioned in the title ‘A Scoping Review with Clinical Advices’. The advices should be brought in the conclusion as a final statement of the authors yet it is missed.
A new brief paragraph with clinical advices has been added before the conclusions (lines 363-401).
Regards
The authors
Reviewer 3 Report
Dear Authors,
I appreciate the importance of the review paper prepared by the team of authors in which they presented the most commonly used tools to assess the status of patients with SABI. The manuscript includes a comparison of the validity of using these scales. However, I would like to point out that in patients who are found to have severe acquired brain injury for various reasons, the assessment of their condition may not necessarily be adequate due to the fact that it is based on behavioral methods. However, for some time now, patients with severe acquired brain injury who are unable to communicate verbally have been assessed using eye-tracking technology. Research indicates (eg. https://doi.org/10.3389/fneur.2022.841095, https://doi.org/10.3390/ijerph19053081) that this method of assessment is not necessarily consistent with traditionally used behavioral methods. I think it would be useful to include this information in the manuscript.
Congratulations on an interesting paper!
Author Response
Thank you for the helpful comments, we have then added the suggested information (lines 350-361, 575-579)
Round 2
Reviewer 2 Report
Thanks to the athours for addressing my concerns. I have no more comments.